# Processing of a Martensitic Tool Steel by Wire-Arc Additive Manufacturing

**DOI:** 10.3390/ma15217408

**Published:** 2022-10-22

**Authors:** Ulf Ziesing, Jonathan Lentz, Arne Röttger, Werner Theisen, Sebastian Weber

**Affiliations:** 1Chair of Materials Technology, Institute for Materials, Ruhr-Universität Bochum, Universitässtraße 150, 44801 Bochum, Germany; 2Chair of New Manufacturing Technologies and Materials, Bergische Universität Wuppertal, Bahnhofstraße 15, 42651 Solingen, Germany

**Keywords:** additive manufacturing, rapid tooling, hot-work tool steel, martensitic steel, wire-arc additive manufacturing, WAAM, DED-Arc

## Abstract

This work investigates the processability of hot-work tool steels by wire-arc additive manufacturing (DED-Arc) from metal-cored wires. The investigations were carried out with the hot-work tool steel X36CrMoWVTi10-3-2. It is shown that a crack-free processing from metal-cored wire is possible, resulting from a low martensite start (M_s_) temperature, high amounts of retained austenite (RA) in combination with increased interpass temperatures during deposition. Overall mechanical properties are similar over the built-up height of 110 mm. High alloying leads to pronounced segregation during processing by DED-Arc, achieving a shift of the secondary hardness maximum towards higher temperatures and higher hardness in as-built + tempered condition in contrast to hardened + tempered condition, which appears to be beneficial for applications of DED-Arc processed material at elevated temperatures.

## 1. Introduction

Additive manufacturing (AM) of metals has steadily gained higher importance over the last decades. In AM a layer-wise build-up facilitates the production of complex-shaped components, which cannot be produced by casting, forging or machining [1]. Additionally, AM allows a high resource efficiency by manufacturing near net-shaped parts and thereby reducing subtractive post-processing, allowing a material utilization of 90 to 100% of the deposited material [2]. Further, AM has the potential to shorten overall lead-times and to simplify supply chains [3].

In particular, AM has gained increasing interest in the toolmaking and tool repair sector, because of the advantages mentioned above [4]. Direct AM of tools is referred to as rapid tooling [5]. Here, AM enables the production of large tools with internal cooling channels, e.g., for hot forming in the automotive sector [6]. With AM, the necessary cooling channels can be created just below the functional surfaces, which enables optimized tool cooling in the process and ultimately shorter cycle times [7]. The result is an increased efficiency of the hot forming process.

Müller has shown that cooling cycles can be reduced by 50% if optimized tool cooling is considered in additively manufactured press-hardening tools in contrast to tools manufactured by casting and machining [8]. In addition to optimal cooling of the tools, a long service life is required, which is achieved through sufficiently high wear and heat resistance [9]. For these applications, tool steels are used. These rely on martensitic hardenability in order to achieve simultaneously high hardness and good toughness [10]. Materials that possess these properties are carbon martensitic hot-work tool steels such as X37CrMoV5-1 (AISI H11) or X40CrMoV5-1 (AISI H13).

However, martensitic transformations often promote cold cracking during processing by various AM processes [11,12]. As a consequence, martensitic steel grades are considered to be technically difficult or impossible to process by AM. On this account, processing of C-martensitic tool steels using AM is focused by several researchers [11,13].

Although a variety of AM processes exist in the metal sector, the AM of metals focusses significantly on the metal powder-based Powder-Bed-Fusion Laser-Beam/Metal (PBF-LB/M) process. Described by Mertens et al., crack-free processing of X40CrMoV5-1 is possible by pre-heating the construction platform above 200 °C [13]. Krell et al. come to similar results, when processing the steel X40CrMoV5-1 using PBF-LB/M [14]. As a disadvantage of this base plate pre-heating, the metal powder tends to oxidize while processing, thus the built components possess a higher amount of non-metallic inclusions, deteriorating the mechanical properties, especially the fatigue properties [15,16]. As a result, the reusability of the metal powder is limited.

Besides the major advantages of high dimensional accuracy and relatively good surface quality (R_a_ = 30 µm), the PBF-LB/M process also has some disadvantages [17]. PBF-LB/M depends on gas-atomized metal powders with a spherical shape as feed stock material, which are expensive to obtain, difficult to handle and not commercially available for many alloys. Furthermore, PBF-LB/M is limited by the size of the powder bed and possesses low deposition rates of 4 to 5 cm^3^/h, thus limiting an efficient manufacturing of complex-shaped large-scale components [1,18]. Additionally, the very localized and focused energy input by laser beam leads to an increased tendency for cold-crack formation, due to the high heating and cooling rates, inducing high thermal tensile stresses [19].

Indeed, many efforts were made to improve the process efficiency by increasing the building rate, several optimization attempts like computational optimization of the design (Design for Additive Manufacturing), the use of multiple lasers, and an improved feeding rate by applying several powder layers and their densification at the same time were developed [20,21].

Despite these improvements, PBF-LB/M remains intrinsically limited in building space, deposition rate and is often more expensive compared to Directed Energy Deposition (DED) AM methods. In DED the most relevant AM processes are Laser Metal Deposition (DED-LB/M) and Wire-Arc Additive Manufacturing (DED-Arc or WAAM). Due to the absence of a powder-bed, these methods have almost no restrictions in size of the manufactured part, which makes them interesting for processing bigger parts [1,2]. Additionally, DED is characterized by its direct and localized deposition of material, which enables so-called hybrid manufacturing, the cost-efficient build-up of material on conventionally processed and pre-shaped substrates.

Deposition rates of up to 1200 cm^3^/h can be achieved by DED in principle, but at the expense of dimensional accuracy and surface quality. For example, feasible deposition rates are ranging from 10 to 200 cm^3^/h (DED-LB/M) and 120 to 500 cm^3^/h (DED-Arc) at mean arithmetical surface roughness R_a_ of >60 µm (DED-LB/M) and >150 µm (DED-Arc) [17,22,23,24]. At this point, it should be mentioned that although a better surface finish is achievable in the PBF-LB/M process, subtractive finishing has often to be carried out on functional surfaces anyways. This means, that there are hardly any differences in the post-processing of the various AM processes, although there are obviously significant differences in the as-built surface quality.

As an interesting and advantageous alternative to other AM processes, in the present study the rapid tooling of tools produced from martensitic tool steels by the DED-Arc process will be characterized. The DED-Arc process facilitates an electric arc for melting the feedstock, i.e., welding wire, and deposits the molten material layer by layer [25]. Here, the principles of gas tungsten arc welding (GTAW) and gas metal arc welding (GMAW) can be applied equally. Whereas, GMAW is preferred for DED-Arc, due to a coaxial feedstock-supply, which simplifies the tool path [23]. Standard welding power sources, torches and wire feeding systems can be used [23]. For moving the torch robotic or computer numerical control (CNC) systems are facilitated for DED-Arc [23]. For making rapid tooling more cost-effective, metal-cored wires, similar to those used for hard-facing, can be applied. These cored wires consist of a mechanically crushed ferro-alloy fill covered by a metallic jacket [26]. The chemical composition can be easily adjusted concerning the composition of the powder particles and their mass fractions, thus, allowing certain flexibility in the chemical composition of the feed-stock, and cored wires are less expensive than metal powders for PBF-LB/M [25,26]. Moreover, the application of wire as a feedstock inserted directly into the heat-source offers high deposition efficiency and counteracts oxidization effects.

In summary, the advantages of DED-Arc lie particularly in the simpler set-up of the manufacturing device in comparison to PBF-LB/M or DED-LB/M, the elimination of work safety measures related to laser safety as well as metal powder handling, the more favorable price of wire-shaped starting material (as an example, wire of the well-known austenitic stainless steel X2CrNiMo17-12-2 (AISI 316L) is about 10 times cheaper than AM powder) and, of course, the freedom of build space and higher building rates.

The processing of martensitic tool steels, such as X37CrMoV5-1 and X40CrMoV5-1 by DED-Arc was investigated recently by several researchers [27,28,29,30]. In summary, these investigations show that DED-Arc can be used to produce defect-free, large-volume specimens from solid wire of hot-work tool steels without external pre-heating. A homogenous hardness of the deposited material can be achieved by keeping the interpass temperatures above the martensite start (M_s_) temperature. From a metallurgical point of view, the high energy input through GMAW in combination with heat accumulation in the solidified material also offers the possibility of realizing increased process temperatures in-situ without using an external heat supply for pre-heating [27,31]. The elevated temperatures could then be used specifically to avoid crack formation in the component and to additively manufacture even high C-alloy steels defect-free.

To our knowledge, DED-Arc processing from metal-cored wires has not been investigated for C-martensitic tool steels, which are alloyed with higher contents of refractory metals than present in X37CrMoV5-1 and X40CrMoV5-1, up to now. Higher alloy contents (i.e., Cr, Mo, W) are necessary in many applications to achieve required application properties and also to improve the processability by additive manufacturing methods /reduce cold cracking tendency. However, the higher alloy contents raise further scientific and technological questions, which will be addressed in the present work:Can chemical homogenous samples be produced by using high alloyed metal-cored wire as feedstock material?What are the microstructures and heat treatment behavior of the specimens processed by DED-Arc without pre-heating?

## 2. Materials and Methods

### 2.1. Materials

For this work the air-hardening hot-work tool steel X36CrMoWVTi10-3-2 was used. This steel originates from repair welding of tools for hot-work applications. The material was provided as metal-cored filler wire DURMAT^®^ FD 818 (DURUM Verschleißschutz GmbH, Willich, Germany) with 1.2 mm diameter. Its nominal chemical composition is given in Table 1. In the following, this steel is referred to as tool steel X36.

For microstructural characterization of the metal-cored wire and the raw materials contained therein, a sample of 10 mm length was cut. The sample was embedded in low viscous Technovit^®^ EPOX (Kulzer GmbH, Wehrheim, Germany) by vacuum assisted infiltration to further allow for standard metallographic preparation (see Section 2.3). Additionally, 5 g of tool steel X36 wire were molten at 1500 °C in an enclosed Al_2_O_3_ crucible in a vacuum induction furnace (500 mbar Ar atmosphere, Leybold-Heraeus, Hanau, Germany) and ground with SiC sand paper to 1000 mesh for following analysis of its actual chemical composition by Glow Discharge Optical Emission Spectroscopy (GDOES) measurements.

### 2.2. Additive Manufacturing of Specimen by DED-Arc

In this work, a circular single layer weld and an additional tubular specimen of 120 layers height (approx. Ø80 mm × 110 mm) were deposited on mild steel substrates (S355, Ø100 mm × 30 mm) according to Figure 1. DED-Arc processing was carried out using a Saprom S3 welding machine (Lorch Schweißtechnik GmbH, Auenwald, Germany) and an automated turntable assembly (TEHAG Maschinenbau GmbH, Bochum, Germany). The welding torch was positioned in a fixture and the substrate was lowered automatically after each revolution by the height of one layer. The metal-cored wire of 1.2 mm diameter was welded under gas shielding (Ar5.0, 15 L/min) using GMAW in direct current electrode positive (DCEP) configuration and standard mode for Fe-base alloys (unpulsed). Prior to this work, a parametric study on single layer welds was carried out, defining the optimal welding parameters in regard to a parabolic-shaped, consistent and spatter-free deposition. For this, voltage, wire feed-rate and welding speed were altered from 17 to 19 V, 3000 to 4000 mm/s and 10 to 16 mm/s respectively, resulting in the parameters given in Table 2 [32]. Additionally, the heat input of DED-Arc of tool steel X36 was derived according to DIN EN 1011-1 [33].

The average bead measures 0.9 mm in height and about 6 mm in width. The duration of one revolution (and thus the interpass time/time until re-welding) in the continuous additive manufacturing of a tubular specimen is 18 s. Temperature measurements were performed during deposition by a thermo couple (type K) attached to the substrate 5 mm below the initial weld layer.

### 2.3. Sample Preparation and Metallographic Preparation

From the single layer weld, a sample of 10 mm width was taken from the center of the weld line and perpendicular to the welding direction.

The processed tubular specimen was separated from the substrate using a wet abrasive cut-off machine Brillant 265 (ATM GmbH, Mammelzen, Germany). Further, slices of 10 mm × 110 mm were extracted. For metallographic investigations these were separated into bottom, mid and top sections (measuring approx. 36 mm) each. Additionally, samples from the specimen’s middle were extracted and cut to approx. 5 mm × 5 mm × 5 mm for following heat treatment.

Specimens for metallographic investigations were ground using standard metallographic procedures by grinding with SiC sand paper from 80 to 1000 mesh, followed by polishing with diamond suspension from grain size 3 to 1 µm and a final polishing step with OPS with a grain size of 0.4 µm. Those for hardness testing were only prepared up to the polishing step with 1 µm diamond suspension.

### 2.4. Heat-Treatment

In order to investigate the influence of DED-Arc on the tempering behavior, tempering was carried out for samples in as-built as well as hardened condition. Therefore, hardening was conducted by austenitizing at 1000 °C (Ar atmosphere +30 min) and subsequent quenching in water. Tempering was performed 2 times for 2 h at temperatures ranging from RT to 650 °C in a convection furnace at ambient air followed by air cooling.

### 2.5. Microscopy

To investigate the microstructure of the deposited material, a field emission scanning electron microscope (SEM) MIRA3 (TESCAN ORSAY HOLDING, a.s, Brno, Czech Republic) was used. The SEM was operated with an acceleration voltage of 15 keV at a working distance of 8 mm in back scattered electron (BSE) contrast mode. For analyzing the chemical composition, energy dispersive spectrometry (EDS) was applied using a X-Max^N^ 50 spectrometer and the corresponding Aztec software (both: Oxford Instruments, High Wycombe Buckinghamshire, UK).

### 2.6. Phase Analysis

For analyzing the present phases in the processed material X-ray diffraction (XRD) was carried out using a D8 Advanced system (Bruker Corporation, Billerica, MA, USA) with a Bragg-Brentano setup. This device is equipped with a Cu X-ray tube (λ = 0.15406 nm). An angular range from 30° to 90° 2θ was investigated at increments of 0.01° 2θ and an exposure time of 5 s.

In order to quantify the local retained austenite (RA) fractions an XRD µ-X360n (Pulstec Industrial Co., Ltd., Nakagawa, Japan) was used. This device is equipped with a Cr X-ray tube (λ = 0.22898 nm) and the measurements were carried out with a collimator of Ø2 mm and an exposure time of 30 s.

### 2.7. Hardness Testing

Vickers hardness tests were performed with an automated testing device KB30S (KB Prüftechnik GmbH, Hochdorf-Assenheim, Germany) in accordance with DIN EN ISO 6507-1 [34]. For each measurement, the testing force was set to 9.807 N (HV1). Along the built-up height, the hardness was measured along the center line every 0.5 mm and was determined by an average of 3 indentations with a horizontal spacing of 1 mm in between. For determination of the tempering behavior, 5 measurements for each temperature were performed and averaged.

### 2.8. Chemical Composition

The chemical composition of the remolten tool steel X36 was measured using a GDOES system of the type GDA 650 HR (SEPCTRUMA Analytik GmbH, Hof, Germany). Four individual measurements with a spot size of 2.5 mm were carried out and averaged. The chemical composition of the DED-Arc-deposited tool steel X36 was derived by the average of six individual measurements with a spot size of 6 mm along the built-up height, applying an optical emission spectrometer (OES) QSG 750 (OBLF GmbH, Witten, Germany).

### 2.9. Thermodynamic Simulations

In order to investigate the segregation behavior during solidification of tool steel X36 processed by DED-Arc the solidification sequence was simulated following the Scheil-Gulliver model. For further investigation of the microstructure evolution of the processed tool steel X36, simulations were carried out for equilibrium conditions. The commercially available software Thermo-Calc-2022a (Thermo-Calc Software AB, Solna, Sweden) with the database TCFE10 was applied. The phases LIQUID, BCC_A2 (ferrite), FCC_A1 (austenite), FCC_A1#2 (MC), MC_ETA (MC), HCP_A3 (M_2_C), M6C_E93 (M_6_C), M7C3_D101 (M_7_C_3_) and M23C6_D84 (M_23_C_6_) were allowed in both simulations. The Scheil-Gulliver simulation was carried out with a substance quantity of 1 mol and a pressure of 1000 mbar, starting at 1600 °C with a step size of 1.0 K, termination condition 99.99% solid phase. The equilibrium simulations were done for the nominal composition and the actual composition of tool steel X36 measured by EDS with a substance quantity of 1 mol and a pressure of 1000 mbar starting from 1500 °C.

## 3. Results and Discussion

The tool steel X36 was chosen for this work due to its similar hardness compared to the common hot-work tool steel X40CrMoV5-1 (AISI H13), but lower tendency for cold cracking in AM processes [27,35]. The lower probability of cold crack formation is achieved by its relatively low martensite start (M_s_) temperature, which is purposefully adjusted by the alloying elements, e.g., Cr, Ni and Mo, according to the so-called low transition temperature (LTT) approach. Further information on LTT-alloys can be found in the works of Zenitani et al. and Kromm et al. [36,37].

### 3.1. Microstructure Formation

#### 3.1.1. Initial State of Material

The initial state of the material is of high importance for the general processability by DED-Arc as well as the microstructure evolution and the material’s overall properties after AM-processing. Thus, the feed stock material was investigated prior to deposition by DED-Arc. Results of the chemical analysis of the wire by GDOES are presented in Table 3. Comparing the results to the nominal chemical composition shown in Table 1 an increased alloying content can be recognized. On the manufacturer’s side, the wires are slightly over-alloyed to compensate for burn-off of elements during the welding process.

The macrostructure of the metal-cored wire is shown in Figure 2a, by a metallographic cross-section. Here a pure iron jacket with a thickness of 100–200 µm can be seen, which is filled with a mixture of pure elements and ferro-alloys. The mechanically crushed particles show irregular shapes (spherical to blocky) and possess a wide size distribution of 2 to around 100 µm. An EDS elemental distribution mapping of the contained particles is shown in Figure 2b. The result shows the presence of the elements Fe (red), Cr (pink), Mn (green) and Mo (cyan) in the individual particles. The particle sizes range from 2–50 µm, whereas FeW (W appears in yellow) particles are typically larger than 100 µm.

Both size and melting temperature of the particles have an influence on the melting behavior of the wire and thus on the achievable homogeneity of the melt pool. Table 4 shows literature values of the liquidus temperatures (T_LIQ_) of the constituents. It can be seen that W, Mo, FeW, FeMo in particular have elevated liquidus temperatures > 1800 °C (>>T_LIQ_-Fe). The energy and time of the process must therefore be selected to be sufficiently high to melt the particles (especially FeW) as completely as possible and to dissolve them uniformly.

#### 3.1.2. Microstructure Formation in a Single Layer Weld

In order to understand the microstructure formation independent of recurring heat input and possible layer-wise re-melting processes, the microstructure of a single weld layer was analyzed in a first step. SEM-images and EDS-element mappings of the as welded-condition of the single layer welds are presented exemplarily in Figure 3 and Figure 4.

The deposited single weld bead of tool steel X36 does not show any sign of crack formation (see Figure 3). The microstructure consists of a mostly martensitic matrix with randomly distributed small carbides and gas pores. As DED-Arc is based on the GMAW process, it is associated with a high cooling rate, i.e., a non-equilibrium solidification. Thus, an increased micro segregation tendency is promoted [44]. Figure 4a shows the SEM images at higher magnification. Here, a solidification structure of primary δ-Fe-dendrites and corresponding micro-segregation in the interdendritic spaces can be recognized. The δ-Fe-dendrites possess a martensitic structure, which originates from an austenitic high temperature phase and high cooling-rates following the welding process. Local differences in the chemical composition are revealed by differences in BSE contrast. Segregations of heavy elements (i.e., Mo, W) can be differentiated by a brighter grey value in comparison to the darker matrix. The EDS measurements (see Table 5 and Figure 4b–j) allow a deeper insight and show that, in fact, every alloying element is present in higher concentrations in the interdendritic regions than in the dendrite cores. It has to be mentioned, that according to the EDS measurements, the chemical composition of the single weld bead (see Table 5) does not comply with the nominal chemical concentration of tool steel X36. Due to the occurrence of fluid motion in the weld pool during arc welding of the dissimilar alloys tool steel X36 and S355, a mixing of the alloying elements is taking place, resulting in a slightly altered chemical composition of the present single layer weld [45].

To further understand the microstructure formation, the solidification sequence was simulated by the Scheil-Gulliver approach. Figure 5a shows the results for the solidification sequence of the tool steel X36 and Figure 5b visualizes the corresponding phase fractions. According to this simulation, primary δ-Fe crystallizes at temperatures below 1465 °C, at 1378 °C the solidification of δ-Fe changes to γ-Fe. At 1373 °C the additional precipitation of MC sets in, followed by the formation of M_6_C below 1255 °C and M_7_C_3_ below 1237 °C. Here, the residual melt solidifies to γ-Fe and MC, M_6_C and M_7_C_3_. Leading to a simulated phase content of 73.67 vol.% δ-Fe and 22.84 vol.% γ-Fe after terminating the solidification at 1201 °C.

Simultaneously to the solidification of δ-Fe and γ-Fe dendrites the residual melt is enriched with all alloying elements in the interdendritic regions. Thus, the corresponding content of the alloying elements, which is dissolved in the δ-Fe and γ-Fe, increases with progress of the solidification process. In this context, Table 6 shows the chemical compositions at the beginning (dendrite cores) and the end of the solidification (interdendritic regions) range of δ-Fe and γ-Fe as well as the corresponding M_s_ temperatures according to Equation (1) by Barbier et al. [46]. Whereas δ-Fe is solidifying almost homogeneously, its overall element concentration is lower than the bulk concentration of tool steel X36. As a result, the γ-Fe is enriched with alloying elements. Thus, M_s_ is lowered, increasing the overall thermodynamic stability of the RA.
M_s_ = 545 − 601.2(1 − exp(−0.868C%)) − 34.4Mn% − 13.7Si% − 9.2Cr% − 17.3Ni% − 15.4Mo% − 2.44Ti% − 361Nb% − 1.4Al% − 16.3Cu% − 3448B% + 10.8V% + 4.7Co%(1)

Subsequently, with the formation of the different carbide types the alloying elements are finally chemically bound, whereby MC carbides solidify Ti- and V-rich, M_6_C Mo- and W-rich and M_7_C_3_ Cr-rich according to the simulation. The local chemical composition along the solidification sequence of each phase can be obtained also by this simulation, showing, that the solidification sequence is affecting the chemical composition of the Fe-matrix phases. These changes in chemical composition show certain similarities to the localized EDS measurements in the dendritic and interdendritic areas (Table 5) mentioned before.

The simulated high fractions of δ-Fe by Scheil-Gulliver approach do not correspond with the experimentally found mostly martensitic microstructure. Therefore, additional thermodynamic equilibrium simulations were carried out. Figure 6a shows a phase quantity diagram of tool steel X36 with nominal chemical composition for thermodynamic equilibrium condition and Figure 6b shows a phase quantity diagram with the actual chemical composition of the single layer weld. Both phase quantity diagrams show major similarities. Here, it becomes evident that a peritectic reaction (L + δ→γ) is taking place in tool steel X36 under equilibrium conditions below 1377 °C. Additionally, the residual δ-Fe is completely transformed to γ-Fe and carbides by a following eutectoid reaction, terminating at 1189 °C.

Under real conditions, the kinetics have to be considered as well. DED-Arc is commonly considered a process with high cooling rates, e.g., up to 60 °K/s, which decreases the peritectic temperatures, resulting in a very fast L + δ→γ transition (initiation and completion within milliseconds) [31,47,48]. As a result, back diffusion of ferrite-stabilizing elements (Cr and Mo for example) into δ-Fe is suppressed. Therefore, the residual melt is enriched with all alloying elements and thus, the interdendritic areas show overall higher element contents (see Table 5).

As a consequence, tool steel X36 transforms to austenite during solidification and subsequent cooling and can eventually transform to martensite by further undercooling. The increased element concentrations in the interdendritic areas affect the M_s_ as shown in Table 6. This finding correlates with the localized EDS-measurements in the single layer weld of tool steel X36, revealing an average M_s_ of 217 °C in the dendritic areas and an average M_s_ of only 136 °C in the interdendritic areas according to Equation (1), indicating, that higher fractions of RA will be in fact concentrated in the interdendritic regions.

#### 3.1.3. Multi-Layer Deposition

The microstructure of the multi-layer specimen is depicted in Figure 7a–c, presenting the bottom (5th layer), mid (60th layer) and top (115th layer) segments of the continuously processed DED-Arc-specimen. Along the built-up height, neither lack of fusion nor signs of crack formation were recognized. Additionally, the deposited material shows a quasi-homogenous chemical composition over the built-up height, presented in Table 7. The microstructures at the investigated positions show certain similarities, consisting of a mostly martensitic matrix with carbides concentrated in the segregated areas and randomly distributed gas pores.

With increasing height, however, the microstructure is coarsening and the carbides’ arrangements are changing towards eutectic networks. Generally, DED-Arc of thin-walled structures is characterized by a particular orientation of heat dissipation, concentrated mostly on the material below the melt pool [49]. The small cross-section of the deposited material therefore reduces the cooling rates to such an extent that heat accumulation of several 100 °C takes place during processing, leading to increased base temperatures for each newly deposited layer [31]. As shown in Figure 7d, heat accumulation of up to 300 °C can be recognized in the substrate, representing the coldest area during processing. Therefore, it can be assumed, that the temperature in the deposited layers is even higher. As a result, the cooling rate of subsequent layers is lowered further, resulting in a slower solidification and therefore affecting the mean dendrite diameter [31]. Thus, the microstructures of a constantly processed tool steel X36 differ from that of a single weld bead on a cold substrate.

By means of XRD (Figure 8), the present phases and fractions thereof were determined. It can be observed that the fractions of RA are increasing while the fractions of carbides are decreasing towards the specimen’s top. The corresponding volume fractions of RA are given in Table 8. Due to the continuous layer-wise deposition, constant re-heating and partial re-melting are taking place in prior deposited and solidified material [50]. Therefore, pronounced diffusion in re-molten as well as solid material occurs, resulting in a decomposition of the enriched austenite [31,49]. Thus, carbide formation is facilitated.

### 3.2. Hardness and Tempering Behavior

For the investigation of the hardness and tempering behavior of DED-Arc-processed tool steel X36, Vickers hardness testing was performed. The very fast cooling at processing a single layer weld of tool steel X36 resulted in an average hardness of 644 ± 7 HV1. The hardness profile of the DED-Arc-processed specimen is presented in Figure 9a. At the specimen’s top, the hardness is similar to that of the single layer weld, whereas the hardness profile is varying over the specimen’s height in a saw-tooth pattern, showing an average hardness of 613 ± 22 HV1. The alternating hardness is attributed to aforementioned multiple re-heating cycles, which cause variations in grain sizes in the heat-affected zones (HAZ), resulting in differences in strength according to the Hall-Petch strengthening mechanism [27]. The resulting microstructure showing multiple HAZ is depicted exemplarily in Figure 9b. The distance in between fusion lines is ranging from 500 to 750 µm, resulting in a varying hardness measured with a step width of 0.5 mm due to indentations in differing regions of HAZ along the specimen height.

More pronounced outliers with lower hardness can be attributed to the presence of gas pores in the deposited material (mean relative density 99.50 ± 0.005%). Known from conventional GMAW of carbon steels, pure Ar shielding gas is not desirable for welding of carbon steels, due to an increased pore formation [51]. Whereas, particular points of high hardness are probably induced by sporadically incompletely molten FeW particles and diffusion-related carbide seams around them, as depicted in Figure 10, which were not investigated further in this work. But, their presence was also mentioned by Großwendt et al. in tool steel X36 processed by PBF-LB/M, resulting from big particle size, high T_LIQ_ and insufficient diffusion during processing [40]. In addition to that, a decrease in hardness of approx. 30 HV1 at the bottom of the specimen (lower 25 mm) is recognized. According to Ali et al., heat dissipation is fast enough in the early stages of deposition to shift the temperature below M_s_, which leads to martensitic transformation in the building process [27]. As shown for the single layer weld, M_s_ of the dendritic areas (according to Scheil-Gulliver simulation and EDS-measurements) are higher than the measured temperatures according to Figure 7d). As a result, martensite is formed in layers close to the bottom in the beginning of the building process (approx. first 5–10 min). In the further course of the building process, the martensite is then partially re-molten, re-austenitized and finally tempered by the heat-input from the deposition of overlying layers, resulting in the presence of tempered martensite, represented by lower hardness values.

With ongoing deposition (>5–10 min), the temperature of the base plate and the build-up part increases above 200–300 °C and thus reaches temperatures above M_s_ (compare Table 6 and Figure 7d). Therefore, the γ→α’ transformation does not take place during the building process and therefore, the transformation in the upper parts sets in after termination of the deposition process. Nevertheless, the presence of carbides in the mid and bottom parts of the specimen in conjunction with the decreasing fractions of RA, shown by XRD (Figure 8 and Table 8), are indicating that time- and diffusion-dependent precipitation of secondary carbides from the austenitic phase, as well as homogenization of microsegregations resulting in carbide formation are also taking place during processing [14,52]. Thus, less carbon can attribute to the lattice distortion during γ→α’ transformation and thus to the hardening of the martensite. As a result, the overall hardness is kept quasi-constant despite the differing microstructural evolution in the deposited specimen.

After performing the post-processing in form of hardening and tempering the hardness was tested again. Figure 11 depicts the influence of tempering on the hardness for as-built + tempered and hardened + tempered condition of DED-Arc processed tool steel X36 in comparison to cast + hardened + tempered condition according to [40]. For low tempering temperatures the hardness of the specimens is equally decreasing due to relaxation effects of the tetragonally distorted martensite and depletion of carbon in the supersaturated martensite [14]. However, the as-built + tempered specimens show a shift of secondary hardness towards elevated temperatures as well as higher hardness reaching 619 ± 5 HV1 at 525 °C. Secondary hardness in tool steels is generally attributed to the precipitation of tempering carbides such as VC and Mo_2_C [10]. As shown by EDS measurements in conjunction with the results of Scheil-Gulliver simulation, the carbide forming elements C, Mo and V show a high degree of segregation and high concentration in the austenite. The high solubility of these elements and slow diffusion in γ-Fe result in a delayed carbide precipitation, thus, transferring the secondary hardness maximum to higher temperatures [14]. This shifting of the secondary hardness maximum appears to be beneficial for application of DED-Arc processed tool steel X36 at elevated temperatures.

## 4. Conclusions

The overall processability of a hot-work tool steel by DED-Arc without pre-heating was investigated in this work. Despite the known probability of cold-cracking of C-martensitic tool steels during GMAW, the hot-work tool steel X36CrMoWVTi10-3-2 was processed fully crack-free. Additionally, the microstructure evolution during DED-Arc applying a metal-cored wire and the effects on hardness and tempering behavior were examined, leading to the following results and conclusions:The LTT-approach is applicable to DED-Arc of C-martensitic steels, achieving a low average M_s_ of 189 °C (according to [46]) and therefore an effective stabilization of RA at low temperatures, resulting in volume fractions of up to 19.0 ± 2.2%;The cooling rates in thin-walled structures are that low that heat-accumulation is taking place, affecting the solidification of each subsequent layer and thus coarsening the microstructure with increasing built-up height;Local chemical inhomogeneities are induced on a macroscopic (incompletely molten W-rich particles) and on a microscopic scale (segregations) during DED-Arc of high alloyed metal-cored wires;A constant stacking of HAZ leads to a saw-tooth patterned hardness over height, local extrema are induced by Ar gas pores and W-rich particles;The secondary hardness maximum of as-built + tempered condition is shifted towards higher tempering temperatures and hardness;

## Figures and Tables

**Figure 1 materials-15-07408-f001:**
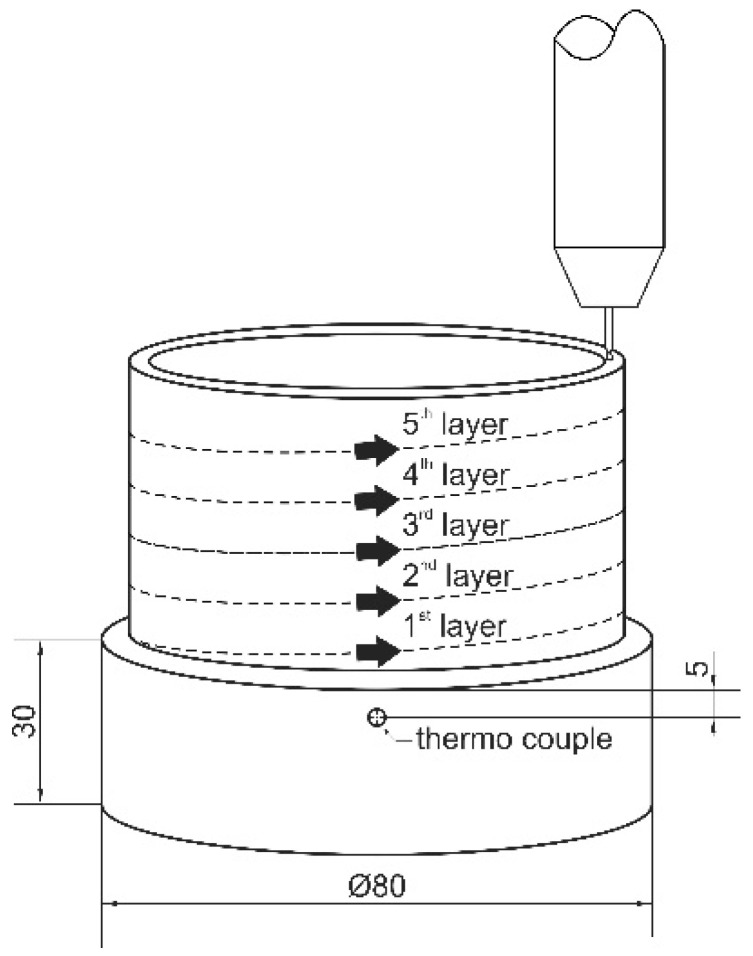
Schematic sketch of the deposition process.

**Figure 2 materials-15-07408-f002:**
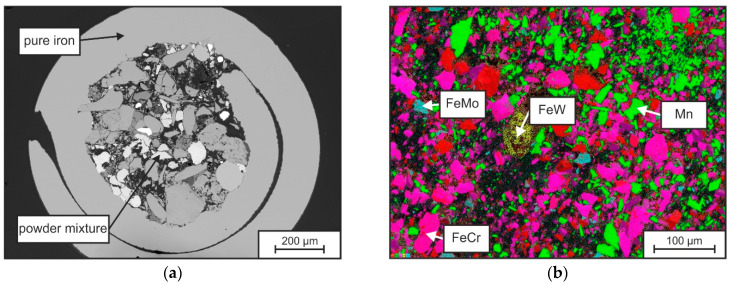
SEM picture of the cross section of the used metal-cored wire (**a**) and element mapping of the contained powder fill (**b**).

**Figure 3 materials-15-07408-f003:**
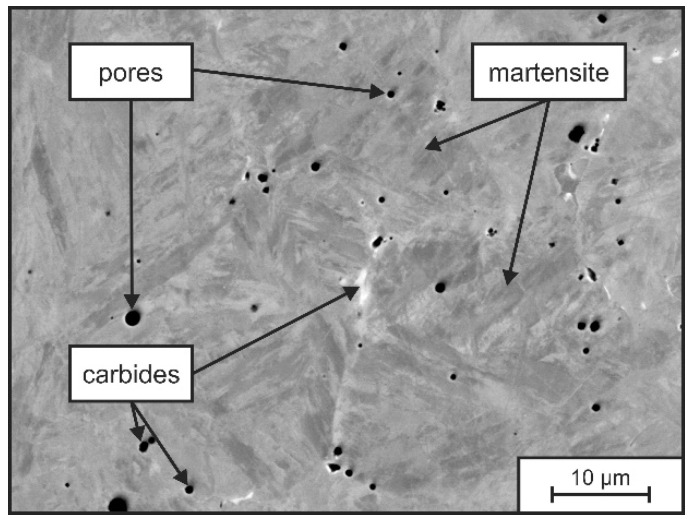
SEM picture of the microstructure of a single weld bead of tool steel X36 in as-welded condition.

**Figure 4 materials-15-07408-f004:**
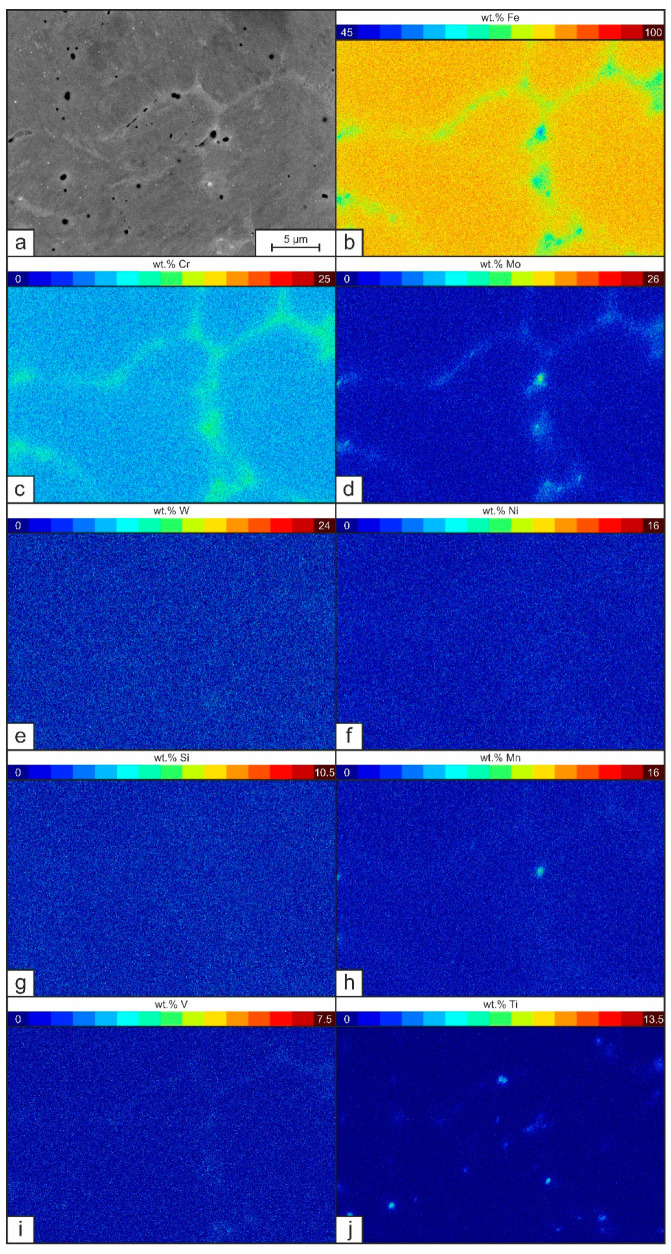
(**a**) SEM picture and EDS element mappings (**b**) Fe, (**c**) Cr, (**d**) Mo, (**e**) W, (**f**) Ni, (**g**) Si, (**h**) Mn, (**i**) V and (**j**) Ti of segregations in a single layer weld of tool steel X36 in as-welded condition.

**Figure 5 materials-15-07408-f005:**
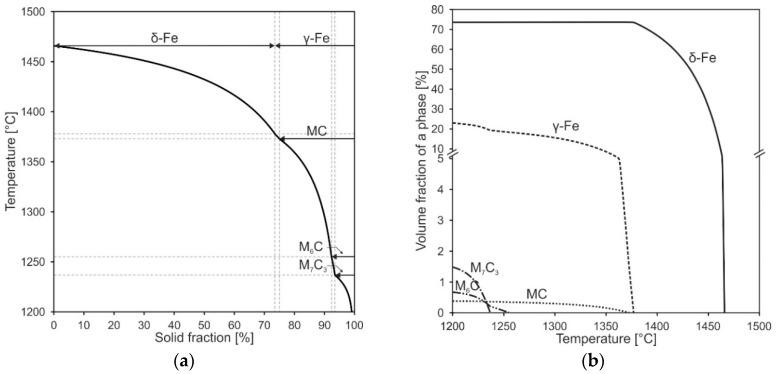
Solidification sequence (**a**) and phase fractions (**b**) of tool steel X36 according to Scheil-Gulliver simulation.

**Figure 6 materials-15-07408-f006:**
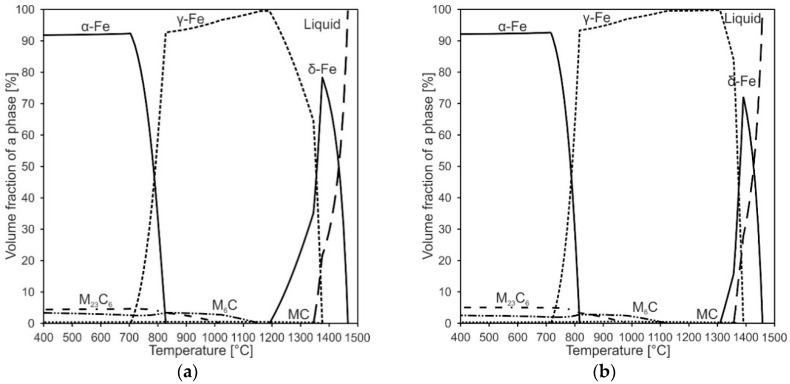
Calculated equilibrium phase fractions of tool steel X36 for nominal chemical composition (**a**) and EDS measured chemical composition (**b**).

**Figure 7 materials-15-07408-f007:**
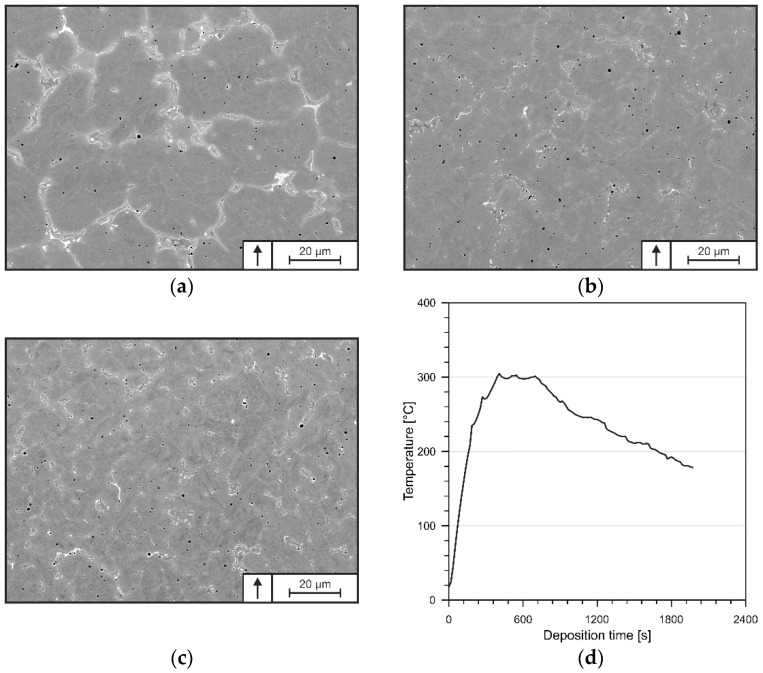
Exemplary SEM pictures of the microstructures at the top (**a**), mid (**b**) and bottom (**c**) of continuously processed tool steel X36 in as-built condition, showing coarsening of the microstructure towards the specimen’s top (**d**) temperature measurements in the substrate during DED-Arc processing of tool steel X36. The arrow marks the build-up direction.

**Figure 8 materials-15-07408-f008:**
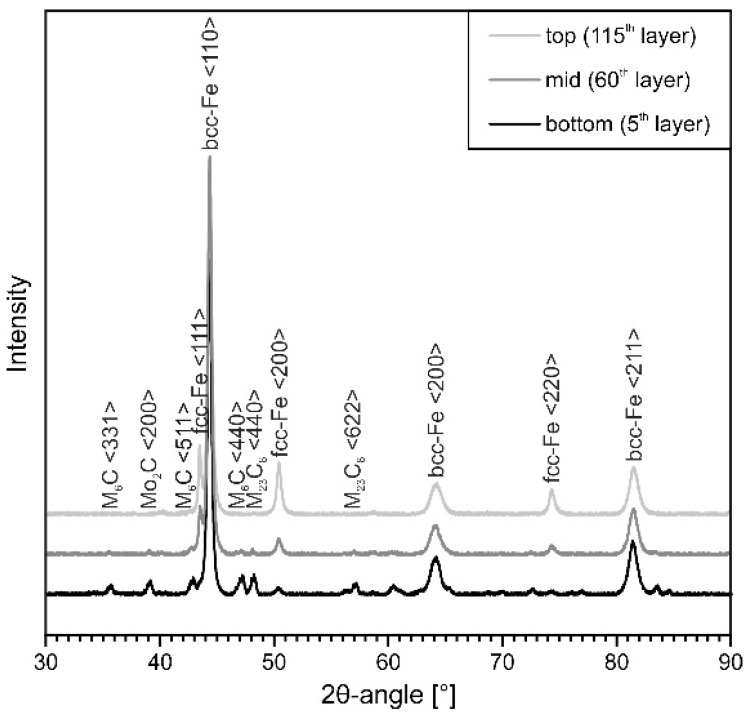
Diffraction patterns at the bottom, mid and top of continuously processed tool steel X36.

**Figure 9 materials-15-07408-f009:**
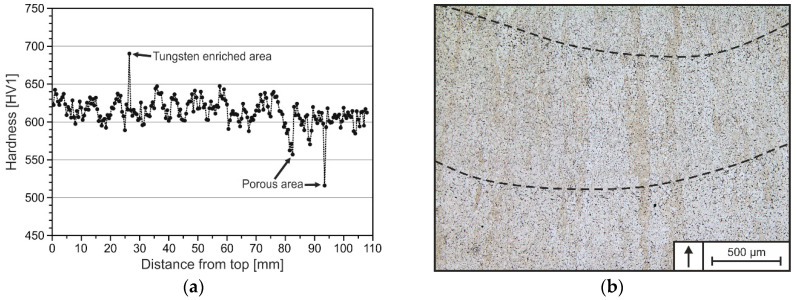
Hardness profile of tool steel X36 over built-up height (**a**) and exemplary picture of multiple HAZ due to continuous deposition (**b**). The fusion lines are indicated by dashed lines and the arrow marks the build-up direction.

**Figure 10 materials-15-07408-f010:**
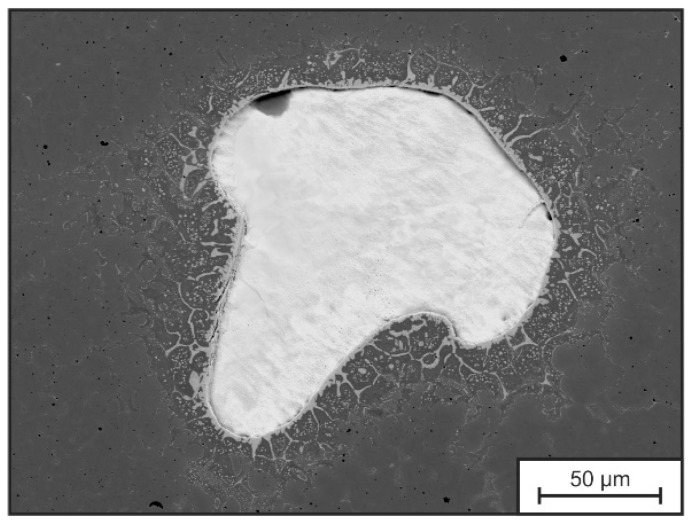
Tool steel X36Exemplary SEM picture of incompletely molten W-rich particle.

**Figure 11 materials-15-07408-f011:**
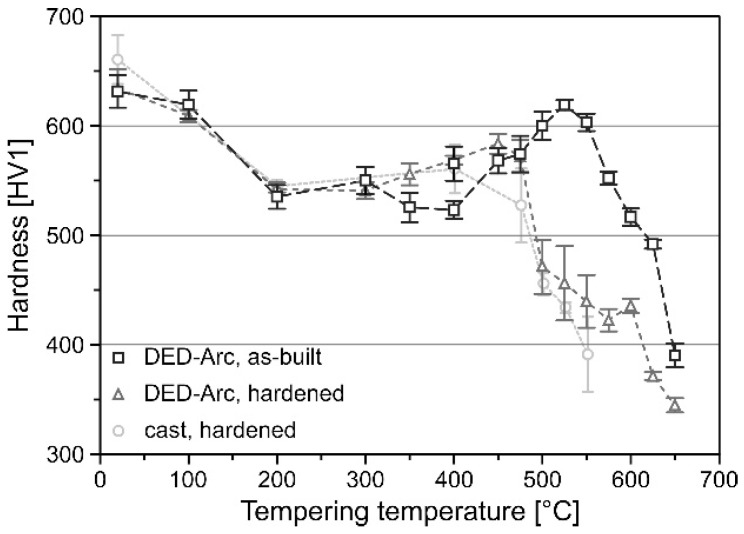
Influence of tempering on the hardness of tool steel X36 processed by DED-Arc and casting according to Großwendt et al. [40].

**Table 1 materials-15-07408-t001:** Nominal chemical composition of the steel X36CrMoWVTi10-3-2 according to standardization.

Element Content (wt.%)									
C	Cr	Mo	W	Ni	Si	Mn	V	Ti	Fe
0.36	10.00	3.00	2.00	1.70	0.70	0.60	0.30	0.20	bal.

**Table 2 materials-15-07408-t002:** Applied GMAW parameters for processing of tool steel X36 specimens.

Torch Angle [°]	Torch Spacing [mm]	Voltage [V]	Amperage [A]	Wire Feed Rate [mm/s]	Welding Speed [mm/s]	Heat Input [J/mm]
90	15	19	~115 ^1^	4000	16	~109

^1^ The amperage was derived automatically by the welding machine based on welding mode, wire diameter and feed rate.

**Table 3 materials-15-07408-t003:** Actual chemical composition of tool steel X36 metal-cored wire.

Element Content (wt.%)									
C	Si	Mn	Cr	Ni	Mo	Ti	V	W	Fe
0.44 ± 0.001	1.31 ± 0.001	1.42 ± 0.001	10.36 ± 0.008	2.18 ± 0.003	3.39 ± 0.002	0.31 ± 0.002	0.38 ± 0.002	2.33 ± 0.026	bal.

**Table 4 materials-15-07408-t004:** Chemical compositions and corresponding T_LIQ_ of typical powder filling particles.

Raw Material	Element Content (wt.%)										T_LIQ_ [°C]
	C	Cr	Mo	W	Si	Mn	V	Ti	Al	Fe	
Fe	-	-	-	-	-	-	-	-	-	99.9	1538 [38]
Cr	-	99.9	-	-	-	-	-	-	-	-	1907 [39]
FeCr HC	7.5	61.2	-	-	2.9	0.3	-	-	-	28.1	1565 [40]
Mo	-	-	99.9	-	-	-	-	-	-	-	2610 [41]
FeMo 70	-	-	67.0	-	-	-	-	-	-	33.0	1841 [40]
W	-	-	-	99.9	-	-	-	-	-	-	3422 [42]
FeW 80	-	-	-	81.0	-	-	-	-	-	19.0	2719 [40]
FeSi 75	-	-	-	-	75.5	-	-	0.1	0.1	24.3	1147 [40]
Mn	99.9	-	-	-	-	-	-	-	-	-	1244 [43]
FeV 80	0.2	-	-	-	0.7	-	80.5	-	1.4	17.2	1657 [40]
FeTi 70	-	-	-	-	-	-	-	70.9	-	29.1	1117 [40]

**Table 5 materials-15-07408-t005:** EDS measurements in the microstructure of a single layer weld of tool steel X36.

Location	Average Element Content (wt.%)									
	C	Si	Mn	Cr	Ni	Mo	Ti	V	W	Fe
global	0.36 ^1^	0.96 ± 0.02	1.03 ± 0.05	8.10 ± 0.05	1.30 ± 0.07	2.31 ± 0.06	0.17 ± 0.03	0.32 ± 0.03	1.92 ± 0.07	bal.
dendrites	0.36 ^1^	0.93 ± 0.03	0.95 ± 0.07	7.90 ± 0.07	1.28 ± 0.10	1.92 ± 0.09	0.05 ± 0.04	0.29 ± 0.04	1.74 ± 0.10	bal.
interdendritic regions	0.36 ^1^	1.02 ± 0.04	1.23 ± 0.08	10.41 ± 0.08	1.42 ± 0.10	4.91 ± 0.10	0.49 ± 0.04	0.53 ± 0.04	2.50 ± 0.10	bal.

^1^ The actual carbon content could not be measured by means of EDS and was therefore set to the nominal value of tool steel X36 of 0.36 wt.%.

**Table 6 materials-15-07408-t006:** Chemical compositions at beginning and end of the solidification of δ-Fe and γ-Fe according to the Scheil-Gulliver simulation.

Phase	Element Content (wt.%)										T_sol_ [°C]	M_s_ [°C]
	C	Cr	Mo	W	Ni	Si	Mn	V	Ti	Fe		
δ-Fe	0.05	9.68	2.67	1.50	1.38	0.63	0.46	0.22	0.05	bal.	1465	343
δ-Fe	0.13	9.68	2.68	1.83	1.89	0.85	0.65	0.25	0.10	bal.	1378	286
γ-Fe	0.33	8.98	2.08	1.35	2.29	0.78	0.72	0.19	0.09	bal.	1378	207
γ-Fe	1.09	6.75	3.37	1.84	1.98	0.06	1.21	0.33	0.03	bal.	1201	−10

**Table 7 materials-15-07408-t007:** Average chemical composition over 120 layers of deposited tool steel X36 measured by OES.

Element Content (wt.%)									
C	Si	Mn	Cr	Ni	Mo	Ti	V	W	Fe
0.36 ± 0.002	1.17 ± 0.019	1.31 ± 0.014	10.07 ± 0.052	1.70 ± 0.013	2.78 ± 0.027	0.17 ± 0.09	0.47 ± 0.015	2.08 ± 0.026	bal.

**Table 8 materials-15-07408-t008:** Volume fractions of RA at bottom, mid and top of continuously processed tool steel X36.

Position	Layer No.	Volume Fractions of Retained Austenite (vol.%)
top	115	19.0 ± 2.2
mid	60	16.0 ± 1.4
bottom	5	12.4 ± 1.9

## Data Availability

The data presented in this study are available on request from the corresponding author.

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
