# Peer review of "Processing of a Martensitic Tool Steel by Wire-Arc Additive Manufacturing"

_materials, 2022, doi:10.3390/ma15217408_

Round 1

Reviewer 1 Report

It is a really interesting study about the welding of tool steel, congratulations.

However the reviewer is not a native English speaking person, the language seems OK.

My concrete remarks, questions, comments are listed in the manuscript_with_reviewers_comments

With proper corrections I think the manuscript can satisfy the publication criteria in Materials.

Reviewer 2 Report

This work describes the processing of hot-work tool steel by wire-arc additive manufacturing. The findings and observations in this work could provide more understanding of materials behaviors during the wire-based AM process, which is reckoned more efficient and cost-effective but less investigated than powder-based AM processes. This manuscript in general is in a good shape, with sufficient analysis of experimental results, supported by some empirical predictions and thermodynamic modelings. After the below points are addressed and a careful check of the language, this manuscript could be accepted and published in the journal Materials.

Section 2.9: Please indicate the termination condition for the scheil caluculation.

The title of Table 6 is suggested as "chemical compositions of δ-Fe and γ-Fe in the dendrite core and interdendritic region...".

Page 11: Using the equilibrium phase diagram to explain the peritectic reaction is not necessary while scheil results can already explain if the full phase names are labeled at each stage of the curve in Figure 5(a).

It would be good to add a micrograph of the melt pool structure at low magnification, showing the shape and dimension of the melt pools, such that it might be compared/correlated to the "saw-tooth" hardness profile.

The tempering response should be compared with the conventionally treated (wrought, solution/quench, temper) counterparts, such that it can then conclude the influence of peak hardness shifting.

Reviewer 3 Report

The authors of this research have tried to use wire arc additive manufacturing to process hot work tool steel. The manuscript is well written and presented and in my opinion it could be considered for possible publication in Materials. However, the manuscript should be slightly revised at some points, addressing comments below before further processing:

-          The English of the manuscript should be improved. There are some grammatical errors and many repeated words which can be easily replaced by other proper ones (ex: page 3 lines 125-131).

-          Put a Ref for section 2.2 (your previous research)

-          Please replace figure 3 with a higher magnification one (higher than that of fig 4). Also attach the microstructure of the martensite to that.

Reviewer 4 Report

This work studied the processability, microstructure and hardness of H11 tool steel fabricated by DED-arc. The paper has provided systematic study on the process-microstructure-property relations upon AM of H11. The microstructure and the experimental data were well documented and reported. The results are of fundamental importance for additive manufacturing of hot-work tool steels. That is being said, this paper can be published after addressing the following comments.

1.    1. The processing parameters was optimised as listed in Table 2. The authors are suggested to provide more details regarding this optimization process. For example, provide optical images to show different densification at different processing parameter set.

2.    2. Volume fractions of retained austenite was provided and it shows significant difference depending on the area of the sample. However, the hardness is homogeneous. The authors need to discuss this point.

3.    3. Hardness cannot fully reflect the mechanical behaviour of the sample. Although the hardness in the sample is homogeneous, the mechanical properties, such as tensile strength, ductility and impact toughness may vary depending on the position of the sample. The statement “Overall mechanical properties are similar over 15 the built-up height of 110 mm.” may be inappropriate. The authors are encouraged to provide more mechanical properties, at least the tensile testing. Otherwise, just use “hardness” rather than “mechanical properties”.

Round 2

Reviewer 4 Report

The authors have addressed the comment for a publication standard.